# The Characterization of a Novel PrMADS11 Transcription Factor from *Pinus radiata* Induced Early in Bent Pine Stem

**DOI:** 10.3390/ijms25137245

**Published:** 2024-06-30

**Authors:** Tamara Méndez, Joselin Guajardo, Nicolás Cruz, Rodrigo A. Gutiérrez, Lorena Norambuena, Andrea Vega, María A. Moya-León, Raúl Herrera

**Affiliations:** 1Instituto de Ciencias Biológicas, Universidad de Talca, Av. Lircay s/n, Talca 3465548, Chile; tamendez@utalca.cl (T.M.); joguajardo@utalca.cl (J.G.); alemoya@utalca.cl (M.A.M.-L.); 2Facultad de Ciencias Agrarias y Forestales, Universidad Técnica Estatal de Quevedo, Quevedo 120313, Ecuador; nicolascruz83@gmail.com; 3Millennium Institute Center for Genome Regulation, Millennium Institute for Integrative Biology, Instituto de Ecología y Biodiversidad, Facultad Ciencias Biológicas, P. Universidad Católica de Chile, Avda, Libertador Bernardo O’Higgins 340, Santiago 8331150, Chile; rgutierrez@bio.puc.cl; 4Plant Molecular Biology Centre, Department of Biology, Facultad de Ciencias, Universidad de Chile, Las Palmeras 3425, Santiago 7750000, Chile; lnorambuena@uchile.cl; 5Facultad de Ingeniería y Ciencias, Universidad Adolfo Ibáñez, Peñalolen 7940000, Chile; andrea.vega@uai.cl

**Keywords:** *Arabidopsis thaliana*, EMSA, MADS-box, MapMan, microarray, radiata pine, response to inclination, STRING, transcription factor

## Abstract

A novel MADS-box transcription factor from *Pinus radiata* D. Don was characterized. *PrMADS11* encodes a protein of 165 amino acids for a MADS-box transcription factor belonging to group II, related to the MIKC protein structure. *PrMADS11* was differentially expressed in the stems of pine trees in response to 45° inclination at early times (1 h)*. Arabidopsis thaliana* was stably transformed with a *35S::PrMADS11* construct in an effort to identify the putative targets of *PrMADS11*. A massive transcriptome analysis revealed 947 differentially expressed genes: 498 genes were up-regulated, and 449 genes were down-regulated due to the over-expression of *PrMADS11*. The gene ontology analysis highlighted a cell wall remodeling function among the differentially expressed genes, suggesting the active participation of cell wall modification required during the response to vertical stem loss. In addition, the phenylpropanoid pathway was also indicated as a *PrMADS11* target, displaying a marked increment in the expression of the genes driven to the biosynthesis of monolignols. The EMSA assays confirmed that PrMADS11 interacts with CArG-box sequences. This TF modulates the gene expression of several molecular pathways, including other TFs, as well as the genes involved in cell wall remodeling. The increment in the lignin content and the genes involved in cell wall dynamics could be an indication of the key role of PrMADS11 in the response to trunk inclination.

## 1. Introduction

The response of trees to vertical loss involves a series of molecular mechanisms that coordinate physiological responses, allowing the recovery of directional growth. Morphological changes in the trunk and the differential accumulation of phenolic compounds have been visualized [1]. The changes in tracheid size and secondary cell wall (SCW) components affect wood quality. The underlying molecular response involves the expression of particular genes, calcium signaling, and the synthesis of hormones such as auxins and ethylene. SCW remodeling is regulated by two large transcription factor (TF) families: R2R3-MYB and NAC [2,3].

MADS-box TFs belong to a highly conserved multigene family previously identified in a wide range of eukaryotic genomes [4]. These proteins are important regulators of plant growth and development, and their genes are expressed in several plant tissues, such as roots, stems, abscission zones, leaves, developing ovules, and embryos [5]. MADS-box TFs can form complexes, and as homo-dimers and hetero-complexes, they can produce a wide diversity of functions [6]. Their function has been mainly described in floral development [7], fruit ripening [8], and anthocyanin biosynthesis [9].

MADS-box TFs share a highly conserved DNA binding domain, called the MADS domain, with 56 to 60 amino acid residues that recognize a conserved [CC(A + T-rich)6GG] motif known as the CArG-box element [10]. MADS-box TFs have been reported in several tree species. In poplar, MADS-box TFs have been reported in vegetative tissues during the differentiation of primary/secondary xylem and phloem and are related to wood formation [11]. In eucalypti, MADS-box TFs have been detected in vegetative tissues [12], and in white spruce, transcripts have been reported in xylem tissue and related to the formation and elongation of vascular cambium cells. Interestingly, in *P. radiata*, nine MADS-box genes have been identified (PrMADS 1 to 9) [13,14]. *PrMADS1* is an ortholog of the AGL2 clade. Meanwhile, *PrMADS2* and *PrMADS3* are members of the AGL6 clade, playing a possible role in the regulation of reproductive development according to Mouradow’s classification [13]. On the other hand, PrMADS 4 to 9 are more abundantly expressed in young flowers than in adult tissues and have been classified in the TM3 MADs-box transcription factor group [14]. Later on, *PrMADS10* was described and reported to be expressed in the stems of one-year-old seedlings in response to inclination, and it was expressed in the lower part of the stem at early times [15]. In the same work, another TF was also mentioned, PrMADS11, but its characterization is still pending. 

The functional analysis of TFs in tree species is not an easy task, so the use of the model plant *Arabidopsis thaliana* can help provide clues about their functional role. For example, the expression of *MADS-box 3* and *4* (*Ptm3/4*) from *Populus tremuloides* in Arabidopsis showed that they are involved in floral development [16], while *Ptm5* participates in vegetative development. The constitutive expression of two members of the Arabidopsis MADS-box family, *AtSHP1* and *AtSHP2* (formerly known as *AtAGL1* and *AtAGL5*, respectively), promoted the lignification of adjacent cells to reproductive organs [17]. 

This article reports the identification and characterization of *PrMADS11*. This MADS-box gene has higher expression levels in bent radiata pine stems, tissues that also rapidly accumulate lignin [18]. Until now, it has been called a truncated gene since it has MADS and keratine-like domains very similar to *PrMADS10* but without a complete C-terminus.

In the present study, the characterization of *PrMADS11* is reported. The role of PrMADS11 on transcriptional regulation is addressed through its over-expression in *A. thaliana* plants. Then, through a microarray assay, a complete transcriptome analysis is performed on *PrMADS11* over-expressing plants, allowing the identification of the genes responding positively (induction) or negatively (repression). Metabolic pathways, which are modulated by the action of this TF, are reported from the analysis of the sets of genes responding to PrMADS11. 

## 2. Results

### 2.1. Characterization of PrMADS11

In previous studies, a partial EST sequence coding for a putative MADS-box sequence was identified at early times when *P. radiata* stems were tilted at 45° [19]. The full-length PrMADS11 cDNA sequence (GenBank accession number KM887511) was obtained using this partial EST sequence as a template, followed by 5′- and 3′-RACE-PCR. With a sequence of 871 bp, an open reading frame (ORF) of 495 bp, and an amino acid sequence of 165 residues, *PrMADS11* codifies for a protein of 19 kDa and pI 6.01. The predicted PrMADS11 protein has the conserved structural features of MIKC type MADS-box TFs, such as MADS and K-box domains, confirming to be a member of the type II MADS-box family, but a short peptide domain at the C-terminal is missing (Figure 1A). 

A phylogenetic analysis was performed with 78 different MADS-box protein sequences, including proteins from *Amborella*, *Arabidopsis*, *Betula*, *Coffea*, *Cryptomeria*, *Eucalyptus*, *Ginkgo*, *Gnetum*, *Malus*, *Oriza*, *Paulownia*, *Picea*, *Pinus*, *Populus*, *Solanum*, *Taxus,* and *Zea* (Figure 1B). The phylogenetic analysis classified PrMADS11 into an SVP-like group according to the MADS sub-classification and very close to orphan genes like AtSVP and StMADS16. The alignment of the sequences from the SVP-like group (Figure 1A) confirmed that they share the MADS-box domain and the K-box domain. Other gene members included in the same cluster are PtMADS5, PrMADS10, PsMADS, OsMADS47, StMADS11, AtAGL24, StMADS16, AmSVP, AtSVP, EgJOINTLESS, PkMADS, PtSVP, GbMADS5, and PgMADS (Figure 1B). 

### 2.2. Differential Gene Expression of PrMADS11 in Young Pine Seedlings

The RNA was extracted from young pine seedling-inclined stems, and the accumulation of *PrMADS11* transcripts was quantified using qRT-PCR in the upper and lower stem sides. High levels of *PrMADS11* transcripts were determined in the lower stem side of the inclined seedlings compared to the control treatment, and the response was very fast as the transcript increment occurred as early as 1 h, especially in lower cut with a big significance (Figure 2) and was maintained after 2 h of treatment although the level of transcripts was lower than the initial measurement, were was prevalent tendence. The accumulation of transcripts at the upper stem side did not change during the experiment.

### 2.3. Gene Modulation in Arabidopsis Plants Over-Expressing PrMADS11

*PrMADS11* was differentially expressed in the lower stem side of inclined pine seedlings, the full-length gene was isolated to transform the Arabidopsis plants, and three transgenic lines over-expressing *PrMADS11* were obtained. A microarray analysis was used to determine the global transcriptome changes resulting from the constitutive over-expression of *PrMADS11*. Nine hundred and forty-seven differentially expressed genes were identified in the transgenic line due to the over-expression of *PrMADS11*. The over-expression of *PrMADS11* induced 498 genes and repressed 449 genes. From those, only 744 genes had protein codes provided by UniProt, DB and furthermore, only 662 were identified within Panther DB and used in the gene ontology analysis (Appendix A). The GO analysis reported that the most representative categories were molecular function and biological process, with 319 and 426 genes, respectively (Figure 3). The main sub-categories within the molecular function category were catalytic activity (48.9% of the genes), binding (27.2%), molecular function as a regulator (9.1%), and transporter activity (9.1%) (Appendix A; Figure 3A). In the case of the biological process category, ten different sub-categories were found, with the most representative being those of cellular processes (36.2%), metabolic processes (26.7%), biological regulation (14.2%), and responses to stimuli (8.8%) (Appendix A; Figure 3B).

The MapMan analysis comprises a set of 34 tree-structured bins describing the central metabolism, as well as other cellular processes (e.g., stress responses). The 166 differentially expressed genes in transgenic Arabidopsis plants were assigned in MapMan terms. The results obtained provided a general view of the different functional and metabolic activities affected by *35S::PrMADS11*, such as major and minor carbohydrates, amino acids, nucleotides, fermentation, lipids, secondary metabolism, and the cell wall (Figure 4A). The genes involved in different metabolic pathways were divided into twenty-seven genes which take part in the modulation of the cell wall, twenty for the metabolism of lipids, fifteen for secondary metabolism, amino acid metabolism (three genes), and others with few genes from the metabolic pathway: the metabolism of light reactions (two genes), major carbohydrates (five genes), minor carbohydrates (three genes), glycolysis (three genes), fermentation (two genes), nucleotide metabolism (seven genes), or mitochondrial electron transport (one gene). When the phenylpropanoid pathway was analyzed, several genes were positively or negatively regulated (Figure 4A).

Four hundred and thirty-nine visible points were determined in the cell function overview (Figure 4B). Three functional groups showed only down-regulated genes: cell division (three genes), the regulation of the cell cycle (six genes), and DNA repair (one gene). The other sixteen different groups showed both up and down-regulated genes. Within the unclassified group, 18 genes were down-regulated, and 26 genes were up-regulated (Figure 4B). Interestingly, the highest number of differentially expressed genes were included within the enzyme family group, where 17 genes were down-regulated, and 34 genes were up-regulated. The second most abundant group was the regulation of transcription, in which 15 genes were down-regulated, and 27 were up-regulated. Other groups with a large number of differentially expressed genes were development, hormones, and stress (biotic and abiotic).

### 2.4. RT-qPCR on Selected Genes

The expression profile of *PrMADS11* was determined for the three transgenic lines (L1-L3), showing that all transgenic lines displayed a higher accumulation of *PrMADS11* transcripts and statistical significance compared to the control and Arabidopsis plants transformed with the empty vector (Figure 5A); notably, the level of transcripts was significantly higher in L1 compared to the other lines. The expression levels of the genes belonging to the phenylpropanoid biosynthesis pathway were examined, considering their involvement in the synthesis of lignin, but also the genes involved in the synthesis of flavonoids, as shown on the right in Figure 5. The three lines over-expressing *PrMADS11* showed an increase in *AtPAL* (Phenylalanine ammonium lyase) transcripts, the first enzyme of the phenylpropanoid pathway, the 3 transgenic lines show a significative relative expression compared to control (Figure 5B). Following the flavonoid biosynthesis pathway, *AtCHS* (Chalcone synthase) displayed an increment in its expression levels, with significant changes in the L1 and L2 lines with *p* < 0.0001 (Figure 5C). Genes such as *AtF3H* (Flavonoid 3-hydroxylase) did not show significant expression changes (Figure 5D), in the case of DFR (Dyhydroflavonol 4-reductase), L2 show more relative expression vs control but L1-L3 had less expression (Figure 5E). In the case of *AtANS* (Anthocyanidin synthase), an enzyme that drives the pathway towards the synthesis of anthocyanins, there was clear repression in the expression of *ANS* in the three lines (Figure 5F). Interestingly, the initial part of the phenylpropanoid pathway is used for the monolignol biosynthesis pathway, and four genes from this route were analyzed: *AtCCR* (Cinnamoyl-CoA reductase), *AtCAD* (Cinnamyl alcohol dehydrogenase), *At*COMT (Caffeic acid 3-O-methyl transferase), and *AtCCoAOMT* (Caffeoyl-CoA O-methyl transferase). There was an increment in the expression of *AtCOMT* in two out of the three lines (Figure 5G), two of the three lines present a inscrease of relative expression but not to significant, in the accumulation of transcripts for *AtCCoAOMT* (Figure 5H), albeit a clear significant increment in the expressions of *AtCCR* (Figure 5I) and *CAD* (Figure 5J). In the case of *FLS* (Flavonol synthase), which drives the pathway toward the biosynthesis of flavonols, two genes were analyzed. Meanwhile, At*FLS1* displayed repression in L3, and in the case of *AtFLS3,* two lines displayed repression (L2 and L3). (Figure 5K,L). Concomitantly, the lignin content was significantly higher in two of the transgenic lines (Figure 5M). The Arabidopsis genome was interrogated, and all the reported genes in this quantitative PCR assay showed response elements in the promotor region, whereas four of them (PAL1, COMT, FLS1, and FLS3) showed more than one cis element (Appendix A). This clearly suggests that the over-expression of *PrMADS11* in Arabidopsis induces the phenylpropanoid pathway, driving the pathway toward the biosynthesis of monolignols and reducing the biosynthesis of anthocyanins.

### 2.5. Protein–Protein Interactions Using STRING

The 744 differentially expressed genes, with protein codes from UniProt, were used in the STRING analysis, but only 526 genes were recognized in the programme database. In addition, the genes without interactions were erased from the analysis. The analysis provided a complex interaction network that could be separated into three clusters. As the molecular pathway involved in the biosynthesis of lignin was the main interest of this research, a zoom into the interactions between TFs and the phenylpropanoid pathway was analyzed (Figure 6). TFs like MYB and NAC were included in one group (*MYB103*, *MYB85*, *MYB83*, *MYB52*, *MYB42*, *NAC010*, *NAC012*, and *NAC073*), as well as genes from the phenylpropanoid pathway (*PAL4* and *4CL4-4CL2*) (amplified view on Figure 6). All showed a type of protein–protein interaction.

### 2.6. Protein–DNA Interaction Analysis using EMSA assays

EMSA assays were performed to determine the interaction of *PrMADS11* with several CArG-box-type sequences. EMSA assays were performed using labeled DNA fragments containing MADS cis elements. Meanwhile, the recombinant *PrMADS11* protein was expressed in yeasts and purified through Ni-NTA affinity chromatography. The results indicate that *PrMADS11* interacts with AG, CARG1, CARG3, and CARG6, as shown by the change in the electrophoretic mobility of the bands (Figure 7A,B). Using unlabeled DNA, the intensity of the lagging band was reduced, indicating competition between the DNA sequences and thus confirming the specificity of the PrMADS11 interaction. The assay also confirmed the absence of an interaction between the *PrMADS11* and MYB sequences, which was used as a negative control (Figure 7B). The purified protein is shown in Figure 7C, and pure *PrMADS11* can be seen in line E.

## 3. Discussion

MADS-box TFs are important regulators of plant growth and development, as well as in the response to environmental changes. Members of this family of proteins have been shown to play a role in floral organogenesis and the regulation of flowering time [20], as well as in the vegetative tissues associated with various plant stresses [21]. Even though some articles have reported the expression of MADS-box TFs in tree stems, few of them have evaluated their role in response to abiotic stress in tree species [15,22,23]. 

*PrMADS11* was differentially expressed in response to stem inclination in one-year-old pine seedlings. Moreover, the accumulation of transcripts was different depending on the stem side, being higher in the lower stem side after 1 h or 2 h of inclination. This TF was initially identified as differentially expressed in an SSH library from young tilted pine seedlings [19]. The gene’s full length is 871 bp long with an ORF of 495 bp and codifies for a deduced protein of 165 amino acid residues. Interestingly, this protein is shorter than PrMADS10, a previously reported MADS-box TF in pine tree [15]. This shorter PrMADS11 protein shares 68.63% identity with PrMADS10 and 78% coverage; both TFs differ at the N-terminal of the sequence near the MADS domain and also at the C-terminal domain. Even if few amino acid residues were missing from the MADS domain, this protein was able to interact with CarG-box-type sequences, indicating that the MADS domain was active, but the missing domain at the C-terminal could influence the modulation of the expression of particular genes. Protein–DNA interactions were confirmed through the EMSA assays, indicating that PrMADS11 can recognize CarG-box sequences in a specific manner. Interestingly, PrMADS10 interacted with the same response elements as PrMADS11, which is expected because both TFs share the same MADS domain. Neither PrMADS11 nor PrMADS10 interacted with a cis element recognized by the MYB TF employed as the negative control in the assay. PrMADS11 shares a similar expression behavior to PrMADS10 in response to vertical loss. However, higher expression levels were quantified in bent radiata pine stems and, importantly, in tissues that rapidly accumulate lignin.

Interestingly, two different MADS proteins have been reported in poplar as specifically expressed in vascular cambium and secondary growth by modulating the homeostasis of auxin [23]. These two MADS TFs are slightly different in size, share a high identity, and both have the main characteristic domains. Most interestingly, the secondary xylem tissue is formed by the redundant action of both TFs, which implies the participation of both TFs in the coordination of the transcription events required during vascular cambium maturation, which drives wood formation. Therefore, the response to inclination in pine trees could be achieved via the action of both the PrMADS10 and PrMADS11 TFs. Different, in this case, is the fact that both genes were expressed at different times during the time course of the bending experiment, with PrMADS11 differentially expressed at an earlier time than PrMADS10 [19]. 

Transgenic Arabidopsis plants over-expressing *PrMADS11* in a constitutive way modified the expression of 947 genes compared to the control plants, with 498 up-regulated and 449 down-regulated genes. It was reported that using the same strategy, *PrMADS10* affected the expression of 1219 differentially expressed genes [15]. The larger number of regulated genes (242 genes) reported for PrMADS10 could be influenced by the missing part at the C-terminal in the shorter PrMADS11 protein. Nevertheless, both TFs modulated the expression of a diverse set of genes involved in many metabolic pathways, and some modulated genes were in common in both over-expressing events. A comparison of the genes differentially expressed in both over-expressing plants focused on TFs, secondary metabolism, and cell wall-related genes is shown in Figure 8. Several MADS-box, MYB, bHLH, and NAC TF types were similarly affected in *PrMADS11* and *PrMADS10* over-expressing plants, but a sub-set of TFs displayed differences between both plants. This is the case of AGL44, MYB90, and MYB103, which were over-expressed in *PrMADS11* over-expressing plants. A set of genes belonging to the phenylpropanoid pathway and driving the biosynthesis of lignin monolignols were equally modified in both over-expressing plants, such as PAL4, 4CL2, 4CL5, and CCoAOMT, suggesting that both TFs take part in the response to abiotic stress. In addition, several genes involved in cell wall structure modification were affected similarly in *PrMADS11* and *PrMADS10* over-expressing plants, being XTH24, FLA11, and CESA8, which were up-regulated, and EXPB3, XTH4/8, and FLA1/2/9, which were down-regulated. In the case of *PrMADS11* over-expressing plants, EXPA15, EXPB1, and XTH15 were down-regulated (Figure 8).

AtPAL4 is connected to At2g28110, a probable glucoroxylan glucuronosyltransferase involved in secondary cell wall biosynthesis, expressed in developing vessels and fiber cells [24] as well as CESA [25]. In addition, both 4CL5 and 4CL2 were up-regulated, as well as in the medicinal plant *Dendrobium huoshanense*. Interestingly, a high FPKM value was reported for the MADS-box in the stem of this species [26]. In turn, these genes were regulated in their expression in transgenic plants and those over-expressing *PrMADS10* [15] or *PrMADS11*.

An increment in the expression levels of CAD, CCR, CCoAOMT, and, in general, COMT, along with a slight increase in lignin, was observed in the *35S::PrMADS11* transgenic lines (Figure 5M). Syringyl units are synthesized when these three genes are simultaneously expressed [27], and it has been noticed that by down-regulating their expression, lignin levels decrease, and the red coloration sign of anthocyanins appears in the xylem [28]. 

Several MYB TFs, such as MYB83/42/52/85/76/50, are modulated in their expression in both transgenic constructions. Functional evidence showed that MYB83 is one of the TFs that activates the SND1 regulating part of secondary wall biosynthesis, and when this gene is over-expressed, a series of cellulose, xylan, and lignin biosynthetic genes are activated, inducing the ectopic deposition of the secondary wall [29] and cell differentiation in Arabidopsis roots [30]. Moreover, MYB83 is specifically expressed in the vascular tissue and findings have shown a positive differential expression. In addition to activating MYB42/52, both are important for producing xylan. On the other hand, MYB76 is a regulator of the biosynthesis of aliphatic glucosinolate and is activated in response to injury, but it was found to have negative values in a microarray analysis [31]. MYB43/32 was differentially expressed in *35S::PrMADS10*, and MYB32 has been described as a repressor of phenylpropanoid biosynthesis, although it specifically regulates pollen wall composition due to being a pollen-specific repressor of lignin biosynthesis [32]. The MYB43 TF affects xylan production [29]. MYB43 was not found to be differentially expressed in the transgenic *35S::PrMADS11* protein; however, the expression of MYB90/103 seemed to be regulated because these were differentially expressed. MYB90 has been associated with the regulation of anthocyanin biosynthesis [33]. At the same time, MYB103 modulates the amount of cellulose via SND1 [29,34]. Interestingly, the accumulation of anthocyanins can be achieved by a depletion in the synthesis of monolignols, with CHS as the key hinge enzyme [35].

In addition, these protein of interest are also associated with the following sub-clade classifications: they are similar to SVP, AGL, and MADS, such as PtSVP (*P. trichocarpa*) and AtSVP (*A. thaliana*), AtAGL24 (*A. thaliana*), StMADS16 (*S. tuberosum*), and PtMADS1 (*P. tomentosa*). PtSVP is a protein from the SVP (short vegetative phase) family and is very closely related to AtSVP, which is a negative floral regulator, and is next to PrMADS10, which is expressed in inclined stems [15]. Most of the genes near PrMADS11 were named orphan genes because StMADS11, StMADS16, AtAGL24, and AtSVP are expressed in vegetative tissues like the vascular cambium region, even though AtAGl24 has been reported as being involved in flowering signals [36,37]. Nevertheless, *PrMADS11* was isolated from the lower pine stem sides shortly after tilting, and its expression was higher in over-expressing plants even if it was truncated, implying a role in the loss of verticality in pine trees in a non-redundant manner. Truncated proteins are molecularly active, as was reported for TMM1, which presents only the MADS domain and plays a role in the response to nitrate in the roots of maize [38]. Interestingly, as in the case reported for PrMADS10, no phenotypic changes were observed in transgenic Arabidopsis plants, which indicates an involvement at the metabolic level. 

On the other hand, more than one member of a family gene can be involved in the construction of a cell wall polymer. Lacasse paralogs were described to be involved in the lignin accumulation of Arabidopsis stems [39]. In this case, five different lacasses with different substrate specificities enable different aromatic substitutions and changing lignin properties regarding the differences in the aliphatic structure of this polymer. The development of the secondary cell wall is a complex process that ended up polymerizing monolignols in a precise spatiotemporal time and allows plants to tolerate biotic and abiotic stresses [39,40].

The presence of several members of a particular gene family and their involvement in a biological process have been described for several genes involved in secondary cell formation [19,38,41]. It has been shown in Arabidopsis that gene duplication causes functional redundancy during the evolution of organisms [42,43]. The truncated *PrMADS11* protein modulates gene expression in vegetative tissues rather than in floral development. The result suggests that the expression of *PrMADS11* is temporally and spatially regulated in pine stem sections [44]. Interestingly, the genes related to the phenylpropanoid pathway and some genes associated with the cell wall were regulated in their expression, including several TFs like MYBs and NACs, which are also involved in wood formation. The non-redundant participation of PrMADS10/11 in response to trunk inclination could be an indication of the response by switching on gradually different target genes. 

## 4. Materials and Methods

### 4.1. Pine Seedlings Subjected to Inclination Stimuli

One-year-old *Pinus radiata* D. Don seedlings that were approximately 30 cm tall were maintained at 20 °C, and nine seedlings were inclined with an angle of 45°. After 1 and 2 h, the stems were cut along the longitudinal axis, providing lower-side and upper-side stem portions as reported previously [19]. The tissue samples were pooled, immediately frozen in liquid nitrogen, and stored at −80 °C until RNA extraction occurred. Additionally, non-inclined seedlings were sampled as a control. 

### 4.2. Gene Cloning and Vector Construction

*Pinus radiata* MADS11 (*PrMADS11*) was cloned from a cDNA library constructed from the stems of young seedlings exposed to inclination at different times [19]. The *PrMADS11* ORF was amplified and cloned following the RACE strategy and using the RNA from inclined stems using SMARTer RACE5‘/3‘kit (Clontech, Mountain View, CA, USA) The PrMADS11-F 5′-CGGACGCGAGTTAACCCTATTTCAAG-3′ and PrMADS11-R 5′-GGACAGCCGAATCTTGAGCCTGAAA-3′ primers were used to obtain the 3′ and 5′ sequences, respectively. The PCR conditions for amplification were 30 cycles each of 1 min at 94 °C, 1 min at 66 °C, and 3 min at 72 °C. A final extension step at 72 °C for 20 min was performed.

*PrMADS11* was cloned into the pBI121 binary vector using forward (5′-TCTAGAATGGCCCGCGAGAAAATAGAAAT-3′) and reverse (5′-GAGCTCATATGGTGGAGGAGATATTA-3′) primers designed to include the XbaI and SacI restriction sites, respectively. The reverse primer had the stop codon deleted. The final construction, denominated as *35S::PrMADS11*, was confirmed via sequencing.

### 4.3. Phylogenetic Analysis

Through a BLASTp analysis and literature search, protein sequences similar to MADS were selected. The listed genes used in the analysis are compiled in Appendix A. The phylogenetic analyses were conducted using MEGA11 [45]. The sequence alignment was performed through the CLUSTAL W method and the phylogenetic tree using the neighbor-joining algorithm with 1000 bootstrap replicates. Multiple sequence alignment was performed with BIOEDIT v7.7 [46] using the CLUSTAL W method.

### 4.4. Stable Transformation of Arabidopsis with Truncated PrMADS11

The Columbia ecotype (Col-0) of *Arabidopsis thaliana* (L.) Heynh plants were transformed using the floral dip method [47]. The seeds from these plants were germinated and placed in vessels containing rock wool and embedded in a hydroponic medium to obtain the T1 population. The plants were maintained at 25 °C in the growth chamber with a long-day photoperiod regime (16 h light/8 h dark). In the T2 generation, the lines were selected based on the segregation of the resistant and sensitive seedlings to glufosinate. Then, three independent lines of the T3 homozygous lines for the *PrMADS11* construct were selected, and the accumulation of the transcripts for the transgene was analyzed using qRT-PCR.

### 4.5. Electrophoretic Mobility Shift Assay (EMSA)

Retardation mobility assays were performed for PrMADS11. For that, the *PrMADS11* coding sequence without the stop codon was amplified employing the following primers: forward (5′-CACCATGGCCCGCGAGAAAATAG-3′) and reverse (5′-ATATGGTGGAGGAGATATTAATTCTTC-3′). The amplified fragment was then ligated from the pENTR SD/D TOPO vector to the pET301/DEST vector using Gateway technology with LR clonase. The construction was amplified in Escherichia coli BL21 cells. Ten colonies were selected and grown in 15 mL of LB growth medium. The expression levels of the proteins were induced by adding 1.6 mM IPTG and grown at 37 °C in a shaker. The proteins were purified from 500 mL of medium, which was centrifuged at 5000 rpm for 5 min at 4 °C. The pellet was suspended in lysis buffer and centrifuged at 10,000× *g*. The protein from the supernatant was purified using a His-tag resin, HisPUR Ni-NTA resin (Thermo Fisher, Rockford, IL, USA) following the protocol indicated by the manufacturer. The protein was quantified using a Bradford reagent.

The EMSA analysis was performed using the Light-Shift Chemiluminescent EMSA kit (Pierce Cat# 20148). The interaction of the PrMADS11 protein was tested with the CArG sequences from AGAMOUS and a series of sequences of different lengths containing similar CArG-box sequences (Appendix A). A cis element recognized by the MYB TF was employed as the negative control. The reactions were carried out in accordance with the manufacturer’s protocol. The sample preparation included recommended volumes of binding buffer, poly (dI·dC), 50% glycerol, 1% NP-40, and 100 mM MgCl_2_. The biotinylated DNA concentrations (20 fmol) were optimized to ensure a linear region for the reaction detection. The nuclear proteins were diluted to an equal total concentration with a final reaction concentration of 120 ug/mL in the DNA mixtures. The samples were run on 6.5% DNA retardation gels and transferred onto 0.45 μm nylon membranes (Amersham Protam Cat# 10600002) for detection on a FujiFilm LAS-3000 CCD camera (FujiFilm, Tokyo, Japon). To confirm the specificity of binding, several experiments were conducted in the presence of unlabeled sequences to ensure changes in chemiluminescence. 

### 4.6. RNA Extraction and Quantitative RT-PCR (RT-qPCR)

The total RNA was extracted from radiata pine seedlings subjected to a vertical loss experiment following the procedure described by Le Provost et al. [48]. The integrity of the RNAs was checked on agarose gels stained with GelRed (Biotium Inc, Fremont, CA, USA), and their concentration was determined using an ND-1000 UV spectrophotometer (Nanodrop Technologies, Montchanin, DE, USA). The cDNA synthesis was performed using a First Strand cDNA Synthesis kit (Fermentas Life Science, Glen Burnie, MD, USA). On the other hand, the total RNA was isolated from *35S::PrMADS11* transgenic Arabidopsis plants using the SV total RNA isolation system (Promega, Fitchburg, MA, USA). 

Primers for the RT-qPCR were designed using Beacon Designer v 2.0 software (Premier Biosoft, Palo Alto, CA, USA). The SYBR Green/ROX quantitative PCR (qPCR) Master Mix (2×; Fermentas Life Science) was used for all qPCR quantifications in a final volume of 20 μL following the manufacturer’s protocol. All experiments were run on a real-time Mx3000P PCR detection system (Stratagene, Cedar Creek, TX, USA) following the protocol described in Cruz et al. [15]. The expression levels were normalized with the stable expression level of three housekeeping genes reported previously [15]. The over-expression of *PrMADS11* was confirmed using 3 biological replicates. The relative expression was calculated using primers for AtFbox, AtUbi10, and AtPP2 as the normalizing genes. The data were analyzed using the methods derived from the algorithm of Vandesompele et al. [49]. The qPCR results were analyzed with a two-way ANOVA with Tukey’s correction post hoc. The differences were considered statistically significant when *p* < 0.05 (*), *p* < 0.01 (**), *p* < 0.001 (***), or *p* < 0.0001(****).

### 4.7. Arabidopsis Promotor Analysis

PlantPAN4.0 was used to search for the MADS cis elements sequences of the MIRK type. The identification of the Tata box was achieved using PlantCARE V1 [50]. The analysis considered the genes related to the phenylpropanoid and flavonoid pathways, and 2000 bp upstream was analyzed. The Arabidopsis genes used in this analysis were PAL, CHS, F3H, DFR, ANS, COMT, CCOAMT, CCR, CAD, FLS1, and FLS3.

### 4.8. Lignin Quantitation

Lignin was determined using the method described by Cruz et al. [15]. The lignin quantification was analyzed using a two-way ANOVA-LSD with Tukey’s correction post hoc, and the significant differences were inferred when *p* ≤ 0.03. 

### 4.9. Microarray Data

The AraGene-1_0-ST 90k chip was used for the analysis, containing 28,501 annotated genes. The raw signal intensity values were first normalized with the RMA method using the affy package in the R language [51,52], and probes were mapped to the locus IDs of the Arabidopsis genome. The raw data were normalized, and differentially expressed genes were determined using standard statistical procedures [53]. The genes with at least a 2-fold change and a *p*-value < 0.05 in the Rank product analysis were considered as differentially expressed. 

### 4.10. Functional Classification Based on MapMan

The gene expression data in a metabolic overview context were visualized using MapMan and the strategy reported by Cruz et al. [15]. The diagram showed positively and negatively regulated genes in red and blue, respectively. The data sets obtained from the microarray analysis (AraGene-1_0-ST) were compared to Ath_AGI_TAIR9_Jan2010 integrated into MapMan from TAIR.

### 4.11. STRING Interaction Network

STRING is a database with known and predicted direct (physical) interactions, as well as indirect (functional) interactions that can be established based on coexpression, colocalization, text mining, or others [54]. All differentially expressed genes with a UniProt code were input, but the genes related to transcription factors and the synthesis of lignin were picked, and the web server was interrogated in order to uncover the potential protein–protein association networks. The database was interrogated for the last time on 31 December (2023).

## 5. Conclusions

The transcription factor PrMADS11 from the radiata pine is expressed in response to stem inclination at an early time (1 h). A high accumulation of *PrMADS11* transcripts was detected in the lower side of inclined pine stems. The functional characterization of this transcription factor was performed by expressing *PrMADS::35S* in *Arabidopsis thaliana*. The over-expression of *PrMADS::35S* modulates the expression of 942 different genes and, interestingly, those involved in cell wall remodeling and transcription factors.

## Figures and Tables

**Figure 1 ijms-25-07245-f001:**
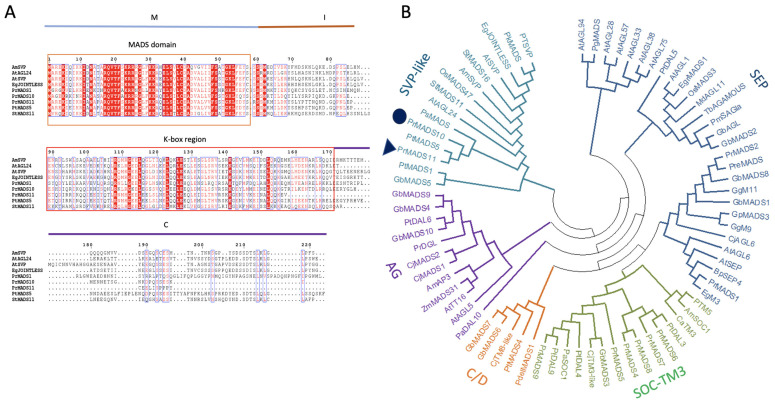
Sequence analysis of the deduced PrMADS11 protein from the radiata pine. (**A**) Multiple alignment of the deduced PrMADS11 sequence with other MADS-box TFs grouped within the same phylogenetic clade. The alignment was performed using Clustal W and ESPript 3.0. The gaps are indicated by dashes, the letters with a red background are identical amino acids, and the letters with a red background are similar amino acids. The blue line indicates the MADS domain, the orange line indicates the interferent site, red line indicates K-box region, and the purple line indicates C-terminal domain. (**B**) The phylogenetic analysis was performed using MEGA 11 software with neighbor-joining, and the bootstrap consensus tree was inferred from 10,000 replicates. The evolutionary distances were computed using the Poisson correction method and are expressed as the number of amino acid substitutions per site. The analysis included 78 amino acid sequences (Appendix A).

**Figure 2 ijms-25-07245-f002:**
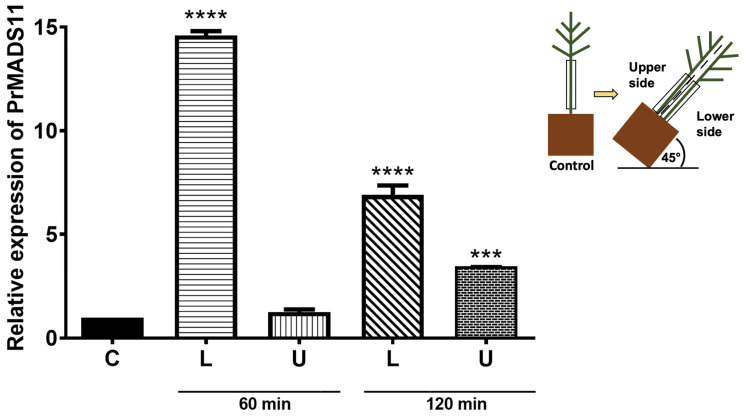
Transcripts levels of *PrMADS11* in young radiata pine seedlings after inclination stimuli. Stem samples were taken at different times of inclination (60 and 120 min), either at the lower stem side (L) or upper stem side (U), as shown on the right of the figure. Non-inclined plants represent control plants (C). The data correspond to the mean ± standard error (SE) of three biological replicates. The asterisks indicate significant differences between the control and inclined samples at each time point and the compare column means the mixed model and multiple comparison test. Tukey’s correction post hoc was used with a 95% confidence interval (*** *p* < 0.001, **** *p* < 0.0001).

**Figure 3 ijms-25-07245-f003:**
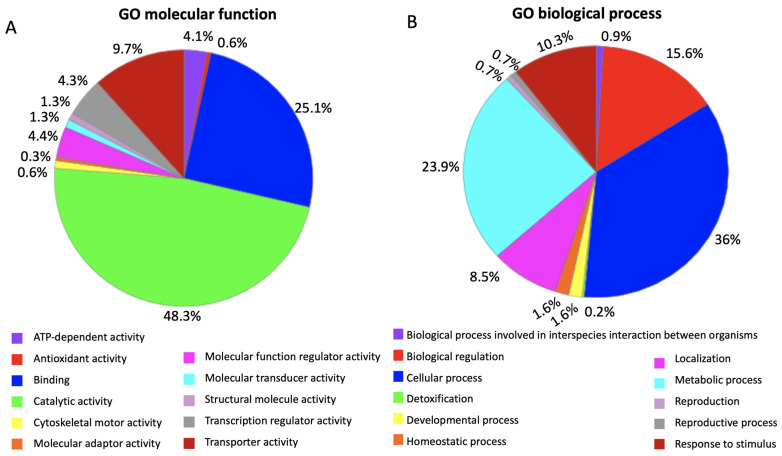
Gene ontology analysis of the differentially expressed genes in bent young radiata pine seedlings. From a total of 744 differentially expressed genes with a UniProt code, 662 genes were identified in Panther DB and used in the GO analysis. (**A**) DEGs are classified according to molecular function and (**B**) according to biological process.

**Figure 4 ijms-25-07245-f004:**
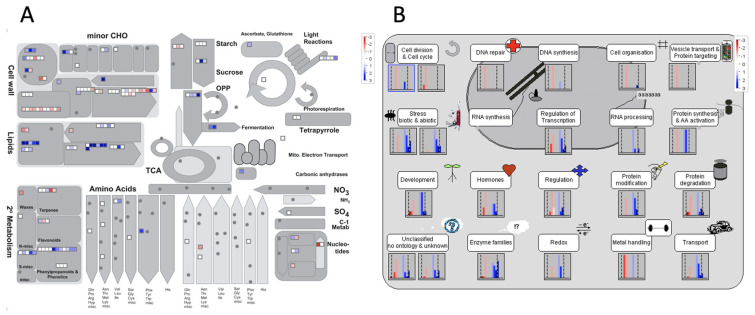
Metabolic overview of DEGs using MapMan. The 1222 differentially expressed genes were used to visualize the changes in the metabolic pathways. One gene could have more than one classification. The up-regulated steps are represented in blue, and the down-regulated steps are in red. (**A**) The metabolic-related genes are further divided as cell wall, lipid, secondary metabolism, amino acid, light, major carbohydrate, minor carbohydrate (minor CHO), glycolysis, and a few for fermentation, the tricarboxylic acid (TCA) cycle, S-assimilation, nucleotide metabolism, tetrapyrrole synthesis, or mitochondrial electron transport. (**B**) Cell function overview. The considered categories were: classification in cell division and cell cycle, DNA repair, DNA synthesis, cell organization, vesicle transport and protein targeting, stress biotic and abiotic, RNA synthesis, regulation of transcription, RNA processing, protein synthesis and AA activation, development, hormones, regulation, protein modification, protein degradation, transport, metal handling, redox, enzyme families and unclassified (no ontology or unknown).

**Figure 5 ijms-25-07245-f005:**
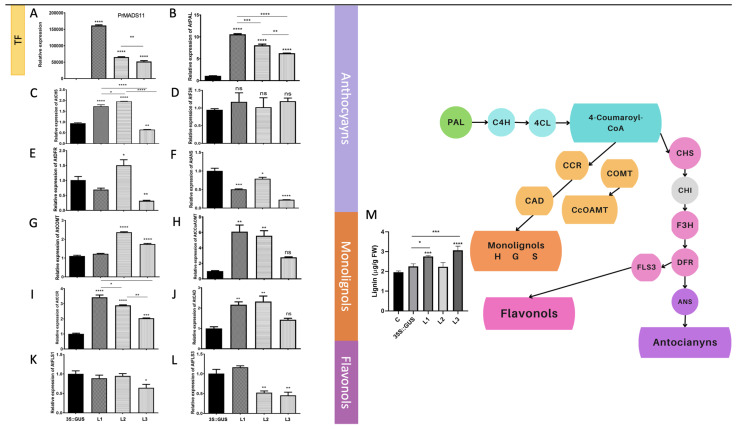
Transcript levels of phenylpropanoid pathway genes in Arabidopsis plants over-expressing the *PrMADS11* gene. The analyzed genes were (**A**) PrMADS11 in transgenic lines, (**B**) *AtPAL* (Phenylalanine amonium lyase), (**C**) *AtCHS* (Chalcone synthase), (**D**) *AtF3H* (Flavonoid 3-hydroxylase), (**E**) At*DFR* (Dyhydroflavonol 4-reductase), (**F**) At*ANS* (Anthocyanidin synthase), (**G**) AtCOMT (Caffeic acid 3-O-methyl transferase), (**H**) At*CCoAOMT* (Caffeoyl-CoA O-methyl transferase), (**I**) At*CCR* (Cinnamoyl-CoA reductase), (**J**) At *CAD* (Cinnamyl alcohol dehydrogenase, (**K**) At*FLS1* (Flavonol synthase), and (**L**) At*FLS3* (Flavonol synthase). The quantitation of lignin was performed in each transgenic line (**M**). The relative expression was determined with RT-qPCR using the housekeeping *AtFbox* (* *p* < 0.05, ** *p* < 0.0014, *** *p* < 0.0002, **** *p* < 0.0001) with Tukey’s test. ns: no significance.

**Figure 6 ijms-25-07245-f006:**
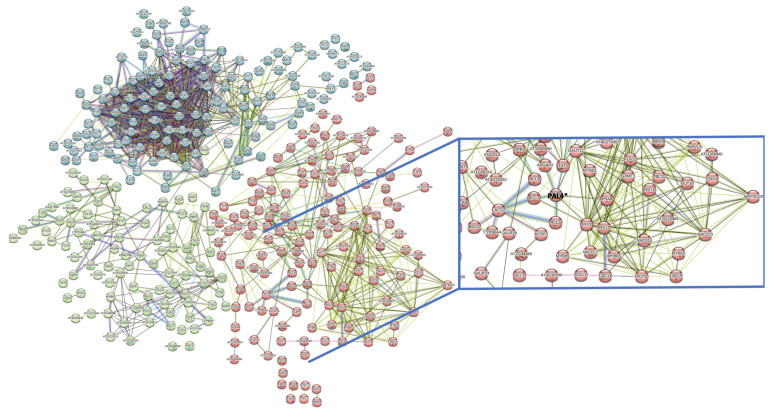
Network of interactions among differentially expressed genes using STRING from the 744 differentially expressed genes identified in Arabidopsis plants over-expressing *PrMADS11*; only those with interactions are shown. The genes were separated into three clusters., cluster 1 in baby blue, cluster 2 in green and cluster 3 in red. Within the rectangle, the interaction between the phenylpropanoid metabolic pathway (PAL4*) and a series of TFs is shown. Each node represents the interaction described in the literature between the proteins.

**Figure 7 ijms-25-07245-f007:**
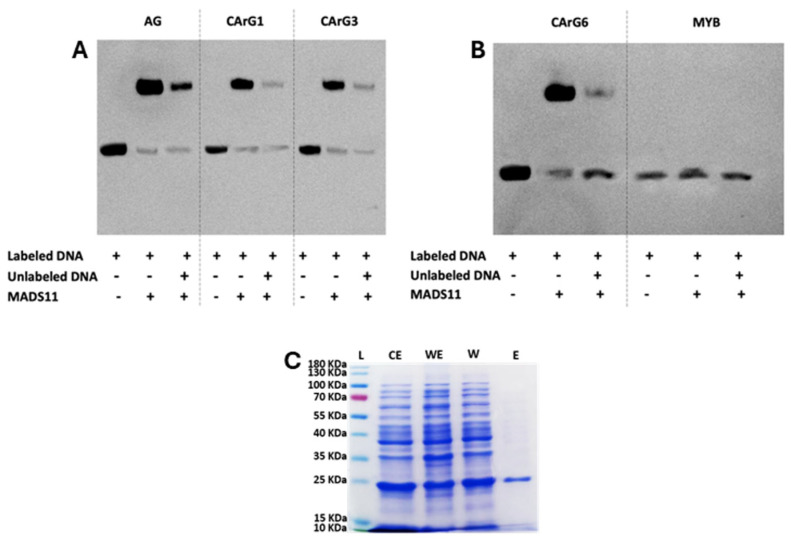
EMSA analysis for PrMADS11. The analysis was performed using the Light-Shift Chemiluminescent kit. The interaction of the PrMADS11 protein was tested against the CArG sequences from AGAMOUS and a series of probes of different lengths containing CArG-box sequences. (**A**) Panel with the probes for Agamus (AG)CArG1, CArG3. (**B**) Panel with the probes for CArG6 and MYB as a negative control. (**C**) PrMADS11 was purified using a His-tag affinity column, and the lines show the ladder (L), crude extract (CE), wash elution (WE), wash (W), and elution (E). The sequences of the CArG probes are presented in Appendix A.

**Figure 8 ijms-25-07245-f008:**
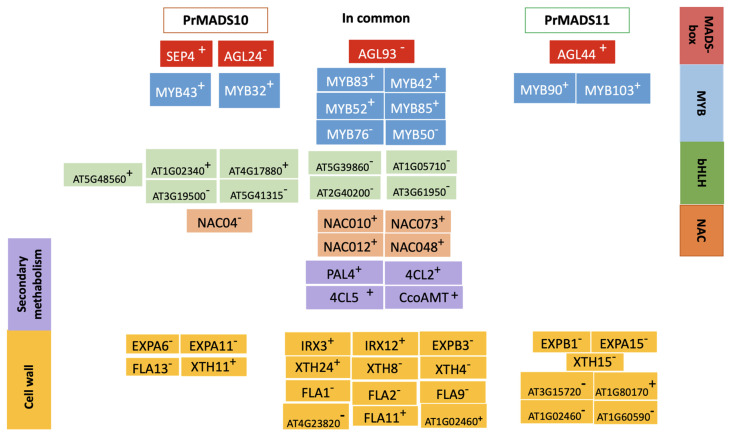
Comparison of the differentially expressed genes between PrMADS10 and PrMADS11 over-expressing Arabidopsis plants. The data for PrMADS10 were obtained from [15]. The comparison was centered on the transcription factors, secondary metabolism, and cell wall. The symbols represent gene activation (+) or gene repression (-).

## Data Availability

Not applicable.

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
