# Peer review of "The Characterization of a Novel PrMADS11 Transcription Factor from Pinus radiata Induced Early in Bent Pine Stem"

_ijms, 2024, doi:10.3390/ijms25137245_

Round 1

Reviewer 1 Report

Comments and Suggestions for Authors

This manuscript demonstrate that one novel MADS-box transcription factor PrMADS11 from Pinus radiata D. Don. PrMADS11 was differentially expressed in the stems of pine trees in response to 45° inclination. PrMADS11 overexpressed Arabidopsis thaliana was obtained. Transcriptome analysis of the PrMADS11 overexpressed Arabidopsis thaliana showed that it was involved in cell wall modification during the response to vertical loss. There are some concerns need to be addressed before publication.

1. The results of the measurement of lignin content in Arabidopsis stems overexpressing PrMADS11Figure S2should be move to the maintext.

2. EMSA results showed that rMADS11 interacts with CArG-box sequences. I think authors should investigate whether those genes whose expression were affected by MADS11 have the CArG-box sequencesIn this way, you may find the potential target gene of MADS11.

3. Authors should discuss the relationship between MADS11 and MADS10. What is the main difference of these two genes?

4. Some of the figures, such as Figure 4, is not clear.

5. There is no citation for figure 8.

6. This manuscript should be carefully read by a native speaker. There are errors. For example, in line 65, “Later on. PrMADS10 was described and reported to be expressed in the stem of one-year-old seedlings in response to inclination, and it was expressed in the lower part of the stem at early times”. Why “later on” is a separate sentence? 

Comments on the Quality of English Language

This manuscript should be carefully read by a native speaker.

Author Response

Dear Editor,

Thanks for your valuable comments.

  1. The results of the measurement of lignin content in Arabidopsis stems overexpressing PrMADS11(Figure S2)should be move to the maintext.

Answer: Thanks for the comment. We have modified figure 5 where transcript analysis of phenylpropanoid and lignin synthesis genes are reported.

  1. EMSA results showed that rMADS11 interacts with CArG-box sequences. I think authors should investigate whether those genes whose expression were affected by MADS11 have the CArG-box sequences?In this way, you may find the potential target gene of MADS11.

Answer: We considered this aspect in the research. In fact, our statement on lines 223-225 say: “Arabidopsis genome was interrogated and all reported genes in this quantitative PCR assay showed response elements in the promotor region, whereas four of them (PAL1, COMT, FLS1, FLS3) showed more than one cis element (Suppl. Fig. 1).” The genes showing differential expression in the transgenic lines do have CArG sequences in their promotor region.

  1. Authors should discuss the relationship between MADS11 and MADS10. What is the main difference of these two genes?

Answer: We considered this aspect in the research. In fact, our statement on lines 285-289. “Interestingly, this protein is shorter than PrMADS10, a previously reported MADS-box TF in pine tree [15]. This shorter PrMADS11 protein shares 68.63% identity with PrMADS10 and 78% coverage; both TFs differ at the N-terminal of the sequence near to the MADS domain, and also at the C-terminal domain.”

  1. Some of the figures, such as Figure 4, is not clear.

Answer: The description of this analysis was re-written.

  1. There is no citation for figure 8.

Answer: We generate figure 8 to facilitate the understading for the effect of over-expressing PrMADS10 and PrMADS11. Those transgenic lines were obtained from different transgenic events and PrMADS10 was reported few years ago. Interestingly, PrMADS11 does not have a portion of the C-terminal but maintain the MADS domain, eventhough is able to modulate several genes in Arabidopsis. The description for this figure can be found in lines 317-319.”A comparison of the genes differentially expressed in both over-expressing plants and focused on TFs, secondary metabolism and cell wall related genes is shown in Figure 8.”

  1. This manuscript should be carefully read by a native speaker. There are errors. For example, in line 65, “Later on. PrMADS10 was described and reported to be expressed in the stem of one-year-old seedlings in response to inclination, and it was expressed in the lower part of the stem at early times”. Why “later on” is a separate sentence? 

Answer: Thanks for the comments the article was reviewed and the English improved. I do regret that the phrase was: “Later on, PrMADS10 was described and reported” and a comma was missing instead of a dot.

Reviewer 2 Report

Comments and Suggestions for Authors

I do not doubt that the methodology is well done, and the manuscript is satisfactory. However, I have some small observations that will contribute to the interpretation of scientific results.

 To avoid empirical comparisons, in paragraph 2.2 "Differential gene expression of PrMADS11 in young pine seedlings" Figure 2, A pairwise comparison like the Tukey test should be used to compare all the treatments. If you want to compare the control vs other treatments, you can also use a Dunnett test. Same for Figure S2.

I found a similar situation in 2.4., instead of using a ttest, I recommend to use the Tukey test to compare all the treatments.

Author Response

 To avoid empirical comparisons, in paragraph 2.2 "Differential gene expression of PrMADS11 in young pine seedlings" Figure 2, A pairwise comparison like the Tukey test should be used to compare all the treatments. If you want to compare the control vs other treatments, you can also use a Dunnett test. Same for Figure S2.

Answer: Thanks for the comment. We performed a new analysis using Tukey test which is incorporated in this version of the article.

I found a similar situation in 2.4., instead of using a ttest, I recommend to use the Tukey test to compare all the treatments.

Answer: Thanks for the comment. We performed a new analysis using Tukey test which is incorporated in this version of the article

Reviewer 3 Report

Comments and Suggestions for Authors

Upon inclination treatment, plants develop a series of molecular responses to alter metabolism and adjust development in order to deal with the stress. In this interesting paper, the authors identified PrMADS11 from radiata pine as a transcription factor whose acute expression is induced by inclination. Overexpression of PrMADS11 in Arabidopsis leads to expression changes of hundreds of genes. Bioinformatic analysis reveals that these genes might activate developmental and molecular pathways for plants to fit the angle change. The findings should be of interest to researchers working on plant development and stress resistance biology. It might be helpful to address several points before publication.

-In Fig.1, does Arabidopsis encode a functional homolog of PrMADS11? In Fig.3, why overexpressed PrMADS11 isolated from radiata pine instead of that from Arabidopsis?

-If WT Arabidopsis is inclined, is the endogenous PrMADS11 homolog overexpressed as what happened in radiata pine?

- Upon PrMADS11 overexpression, expression of some genes got upregulated while others down-regulated. In radiata pine, does their expression under the inclination treatment display a similar pattern as what has been shown in Arabidopsis (overexpressing PrMADS11)?

-For Fig.7, it is better to show a PAGE gel with purified proteins.

Comments on the Quality of English Language

- English writing should be improved as there are many grammar errors in the paper, of which I only picked up some as follows:

PrMADS11 encodes a protein of 871 bp (165 amino acids, a protein has …aa while a gene has …bp) for a MADS-box transcription factor belonging to group II.(of what?)

The increment in lignin content 30 and those gene(s) involved in cell wall dynamics

High levels of PrMADS11 transcripts were determined in the lower stem side of 121 inclined seedlings compare (compared) to control treatment

C (control) means non inclined (non-inclined) seedlings

Data correspond to mean 129 ± SE (standard error? Needs a full name for the 1st time) of three biological replicates.

BIOEDIT {Citation, a real citation}using the CLUSTAL W method.

For that, PrMADS11 codying (coding) sequence

The expression level of genes belonging to the phenylpropanoid biosynthesis pathway was performed (examined)

of different lenghts (lengths) containing CArG-box sequences

Even though,(delete the comma) some articles have reported

Interestingly, PrMADS interact (interacts) with the same

A set of genes belonging to the phenylpropanoid 310 pathway and driven (driving on) the road towards the biosynthesis

Author Response

-In Fig.1, does Arabidopsis encode a functional homolog of PrMADS11? In Fig.3, why overexpressed PrMADS11 isolated from radiata pine instead of that from Arabidopsis?

Answer: The phylogenetic analysis showed that PrMADS11 belong to the same general cluster as AtAGL24 and AtSVP but is not in the same clade. The closest protein from Arabidopsis has 51,2% identity and a coverture of 92% with PrMADS1. Additionally, several isoforms in the case of AtSVP can be found. A functional analysis should be performed in order to confirm that the Arabidopsis protein has the same function as the pine MADS11, but the evidence indicate that AGL24 is involved in flowering, even if the gene transcript has been reported in stems (Yu et al., 2002).

In Fig.3, why overexpressed PrMADS11 isolated from radiata pine instead of that from Arabidopsis?

Answer: The first aim of the work was to identify differentially expressed genes in response to pine inclination (Ramos et al., 2012). We reported several genes in that article, but others after that work were deeply studied. As a way to make a functional characterization of the role for PrMADS11 we choose as a reference model A. thaliana. Another consideration was the fact that a reference genome is annotated and available for this specie but also, having in mind the difficulty to transform radiata pine. At the same time, it is far easier to obtain homozygous T3 from A. thaliana which takes a shorter time. Seeds germination and young plants can be obtained faster which is another advantage of using Arabidopsis as model specie for this experiment.

-If WT Arabidopsis is inclined, is the endogenous PrMADS11 homolog overexpressed as what happened in radiata pine?

Answer: MADS11 belong to radiata pine and there is no ortholog in Arabidopsis. The phylogenetic analysis indicates that two genes from arabadopsis grouped in the same phylogenetic branch, but in a different clade. Moreover, functional characterization of AtAGL24 indicate that is related to flowering, eventhough, the accumulation of transcript has been reported in shoots. In this sense, we can not determine the expression profile of PrMADS11 in inclined arabidopsis plants.

- Upon PrMADS11 overexpression, expression of some genes got upregulated while others down-regulated. In radiata pine, does their expression under the inclination treatment display a similar pattern as what has been shown in Arabidopsis (overexpressing PrMADS11)?

Answer: Arabidopsis was used to be transformed due to short time in generating offsprings. The transformation of radiata pine is not an easy task and takes long in getting transgenic plants, making difficult to observe phenotypic changes. The PrMADS11 was identified in our previous work were several genes associated to the synthesis of lignin were also identified, which were also differentially expressed as in Arabidopsis (Ramos et al., 2012). The information was introduced in the discussion.

-For Fig.7, it is better to show a PAGE gel with purified proteins.

Answer: The result on the protein separation was incorporated into the article.

Comments on the Quality of English Language

- English writing should be improved as there are many grammar errors in the paper, of which I only picked up some as follows:

PrMADS11 encodes a protein of 871 bp (165 amino acids, a protein has …aa while a gene has …bp) for a MADS-box transcription factor belonging to group II.(of what?)

Answer: Introduction was modified.

The increment in lignin content 30 and those gene(s) involved in cell wall dynamics

High levels of PrMADS11 transcripts were determined in the lower stem side of 121 inclined seedlings compare (compared) to control treatment

C (control) means non inclined (non-inclined) seedlings

Data correspond to mean 129 ± SE (standard error? Needs a full name for the 1st time) of three biological replicates.

BIOEDIT {Citation, a real citation}using the CLUSTAL W method.

Answer: New information was incorporate regarding the Bioedit version and reference.

For that, PrMADS11 codying (coding) sequence

The expression level of genes belonging to the phenylpropanoid biosynthesis pathway was performed (examined)

of different lenghts (lengths) containing CArG-box sequences

Even though,(delete the comma) some articles have reported

Interestingly, PrMADS interact (interacts) with the same

A set of genes belonging to the phenylpropanoid 310 pathway and driven (driving on) the road towards the biosynthesis

Answer: Thanks for the comments on the grammatical errors. The article was reviewed and the English improved.

Round 2

Reviewer 1 Report

Comments and Suggestions for Authors

Some figures are blurry.

Please provide figures with higher  resolution. 

Author Response

Dear Reviewer,

Thanks for your comment. We have provided new figure with a better quality and png extension.